# Entropy-Based Measures of Hypnopompic Heart Rate Variability Contribute to the Automatic Prediction of Cardiovascular Events

**DOI:** 10.3390/e22020241

**Published:** 2020-02-20

**Authors:** Xueya Yan, Lulu Zhang, Jinlian Li, Ding Du, Fengzhen Hou

**Affiliations:** 1School of Science, China Pharmaceutical University, Nanjing 210009, China; xueya_yan@163.com (X.Y.); zhang_lulu@stu.cpu.edu.cn (L.Z.); ddmn9999@cpu.edu.cn (D.D.); 2School of environment science, Nanjing Xiaozhuang University, Nanjing 211171, China; alian1294@njxzc.edu.cn

**Keywords:** heart rate variability, cardiovascular disease, sleep, machine learning, XGBoost

## Abstract

Surges in sympathetic activity should be a major contributor to the frequent occurrence of cardiovascular events towards the end of nocturnal sleep. We aimed to investigate whether the analysis of hypnopompic heart rate variability (HRV) could assist in the prediction of cardiovascular disease (CVD). 2217 baseline CVD-free subjects were identified and divided into CVD group and non-CVD group, according to the presence of CVD during a follow-up visit. HRV measures derived from time domain analysis, frequency domain analysis and nonlinear analysis were employed to characterize cardiac functioning. Machine learning models for both long-term and short-term CVD prediction were then constructed, based on hypnopompic HRV metrics and other typical CVD risk factors. CVD was associated with significant alterations in hypnopompic HRV. An accuracy of 81.4% was achieved in short-term prediction of CVD, demonstrating a 10.7% increase compared with long-term prediction. There was a decline of more than 6% in the predictive performance of short-term CVD outcomes without HRV metrics. The complexity of hypnopompic HRV, measured by entropy-based indices, contributed considerably to the prediction and achieved greater importance in the proposed models than conventional HRV measures. Our findings suggest that Hypnopompic HRV assists the prediction of CVD outcomes, especially the occurrence of CVD event within two years.

## 1. Introduction

Cardiovascular disease (CVD) is one of the major causes of mortality in the world, which has seriously influenced sustainable human development [1]. In 2015, the number of CVD patients was up to 422.7 million and 17.92 million people died of CVD all over the world [2]. Screening in high risk population and early intervention of CVD are highly recommended in clinical practice. In the past decades, a lot of efforts from the medical community have been devoted to the prediction and evaluation of potential CVD, but the challenges remain unsolved.

Evidence in scientific literature clearly demonstrated an uneven distribution of the occurrence of cardiovascular accidents during the 24 h [3]. CVD events have the tendency to occur at the end of the sleep period and during wakefulness after sleep, instead of later in the day [4]. Although it’s still unclear what is the mechanism responsible for the increased occurrence of cardiovascular events in the morning, surges in sympathetic activity are generally considered as a primary cause [4,5,6,7,8,9]. As a non-invasive way, heart rate variability (HRV) becomes the most commonly used method to evaluate the function of the autonomic nerve system (ANS) [10,11]. HRV conventionally includes time domain and frequency domain analysis [12]. In recent years, with increased awareness of complexity theories, entropy-based measures have been included as part of the nonlinear analysis of HRV, or as an independent part in addition to conventional HRV measures. An increasing number of studies confirmed the fact that healthy heart rate rhythms exhibit features of complex systems reflecting dynamic, non-stationary and nonlinear properties [13,14].

HRV is a widespread used method to assess the degree to which autonomic control of the heart is impaired [15]. Numerous studies indicate that HRV indices could be used as prognostic markers for an extensive range of cardiovascular disorders [15]. The range can vary from less lethal, such as hypertension and uncomplicated diabetes mellitus to the conditions of myocardial infarction and heart failure [15]. For example, time domain and frequency domain HRV indices have been suggested as potential predictors of cardiac mortality after myocardial infarction (MI) [16,17], and compared with other traditional estimations, and they provided a better prediction of death caused by progressive heart failure [18]. As for the entropy-based measures, Yi-Lwun Ho et al. found that multiscale sample entropy (MSE) analysis of HRV could potentially offer an alternative marker for the outcome of congestive heart failure (CHF) in addition to neurohormonal inhibition therapy [19], and Eiichi Watanabe et al. demonstrated that MSE provided independent prediction of ischemic stroke [20]. Studies also have shown that another entropy measure of HRV, permutation entropy (PE), was a helpful metric in the assessment of patients suffering from cardiodepressive vasovagal syncope [21], and it played a better role in the evaluation of cardiovascular autonomic neuropathy than other traditional methods [22].

However, in the existing systems of cardiovascular risk evaluation, such as QRISK2 risk score [23], and Framingham risk score [24], HRV characteristics haven’t been taken into consideration. Only typical risk factors of CVD, such as age, blood pressure, smoking, diabetes and hypertension, are considered to obtain a prediction of CVD. With the development of artificial intelligence in computer science, evidence has shown that machine learning models which employed HRV indices as features performed more accurately than traditional modified early warning score in the forecast of cardiac arrest [25]. Furthermore, our recently work suggested that machine learning models, combining some typical CVD risk factors and the conventional time domain and frequency domain indices of sleep HRV, is promising in the early prediction of CVD events [26], such as hospitalized acute myocardial infarction, coronary surgical intervention, congestive heart failure and any other heart and cardiovascular-related diseases.

Therefore, in this study, we aimed to do a pilot testing to explore the potential value of hypnopompic HRV (i.e., the sleep HRV preceding morning awakening), especially its complexity, in the automatic prediction of CVD events. We identified a group of baseline CVD-free subjects in an open database, and further divided them into two groups, CVD group and non-CVD group, according to the presence of CVD during follow-up years. Meanwhile, we aimed to investigate whether hypnopompic HRV had a better ability for the prediction of short-term CVD events than long-term outcomes. Thus, a subgroup of the CVD group was established with subjects who had a CVD event within two years after baseline. Machine learning models for both long-term and short-term CVD prediction were constructed, based on the typical CVD risk factors and HRV characteristics. Model performance and feature importance were further evaluated and compared.

## 2. Materials and Methods 

### 2.1. Participants

The data used in the present study were obtained from an open access database, the Sleep Heart Health Study (SHHS) [27], which is a multi-center prospective cohort study implemented by the National Heart Lung and Blood Institute to determine whether sleep-disordered breathing is a risk factor for CVDs. Between November 1, 1995, and January 31, 1998, a total of 6441 individuals aged 40 years or older were enrolled in the baseline polysomnogram (PSG) collection using the Compumedics SleepWatch polysomnograph at home. Follow-up visits were conducted until 2011 to record the occurrence and latency of CVD events, such as hospitalized acute myocardial infarction, coronary surgical intervention and CHF.

In the present study, 2442 participants were included as they: (1) Were CVD-free at baseline; (2) had an acceptable sleep efficiency (higher than mean minus double of standard deviation of all subjects) during the night of PSG recording; (3) didn’t take benzodiazepines, tricyclic or non-tricyclic antidepressants within two weeks; (4) didn’t drink beer, wine or coffee within 4 h before going to bed; (5) and their electrocardiogram (ECG) data and basic clinical characteristics were available. The included participants were divided into two groups, CVD group and non-CVD group, according to whether they experienced CVD incidents between baseline and the end of the follow-up visit (2011). As a result, there were 1359 participants in the CVD group and 1083 participants in non-CVD group.

### 2.2. Signal Processing

The ECG data were recorded at a sampling rate of 125 Hz and with a Butterworth band-pass filter (0.5–45 Hz). For each participant, one-hour ECG segment before morning awaking was used. According to Pan-Tompkins’s method, every R-wave peak of the ECG segments was detected [28], then the intervals between two successive R-wave peaks were calculated and denoted as RR intervals in the following. Interpolation of RR intervals was adopted to reduce the bias introduced by ectopic beats or other artifacts [29]. The participants whose RR intervals had a percent of artifacts or ectopic beats more than 20% were excluded from further analysis, resulting in 1219 and 998 participants in the CVD group and non-CVD group, respectively. Out of the CVD group, 70 participants had a CVD event within two years after baseline.

### 2.3. HRV Analyses

In this study, for each participant, HRV were measured in two kinds of way. In one hand, the conventional frequency-domain analysis was first employed on each 5-min HRV segment [12], and then averaged on all 12 segments to get the short-term characteristics of HRV. On the other hand, time-domain analysis and nonlinear dynamic analysis were applied to the whole one-hour data for an evaluation of long-term HRV. The used HRV measures were summarized in Table 1. 

MSE was introduced to quantify the complexity of biologic systems [30,33,34]. Two steps were required in calculating MSE of a RR interval series in the present study, coarse-graining the original series according to a scale factor [30], and evaluating the sample entropy of each coarse-grained time series [35]. Here, the scale factor was set as 10, resulting in 10 values of sample entropy which were sequentially denoted as MSE1, MSE2, etc., in the following. For the calculation of sample entropy, the embedding dimension m was set as 2 and the tolerance r as 0.15 × SD, where SD is the standard deviation of the original time series. 

As far as the PE algorithm is concerned, the original approach proposed by Bandt and Pompe [31] assumes that the time series has a continuous distribution and simply ranks equal values in the vector according to their order of emergence. However, sometimes the results can be biased by the un-trivial amount of equal values in a time series, due to a limited sampling rate. Bian et al. [32], thus, proposed the modified permutation entropy (MPE) method and demonstrated that it could achieve a better characterization of the cardiac system states than the original PE [32]. In the MPE algorithm, equal values in a vector are converted to equal symbols in the permutation pattern. Thus, in the present study, the MPE of RR intervals was utilized instead of the original PE algorithm. In the calculation of MPE, the embedding dimension *m* was set as 4.

### 2.4. Automatic Prediction of CVD Outcomes

In this study, HRV metrics and other clinical characteristics, i.e., age, gender, body mass index (BMI), height, waist/hip ratio, smoking status, lifetime cigarette smoke, diabetes, hypertension, apnea hypopnea index (AHI) and respiratory disturbance index (RDI), were used to predict whether the participant would have: (1) A long-term cardiovascular event within the following average 11.2 years (Q1–Q3, 11.1–12.5 years) or (2) a short-term cardiovascular event within two years. The extreme gradient boost (XGBoost) algorithm was adopted to construct the machine learning model.

#### 2.4.1. A Brief Introduction to XGBoost Algorithm

XGBoost, proposed by Tianqi Chen [36], is an improved boosting algorithm based on gradient boosting decision tree. XGBoost is composed of many classification and regression trees (CARTs), and the final predictive result of each sample is the sum of all CARTs. Each CART is formed by numerous times of splits, and for each split of a node, we can calculate the gain before and after split. A higher value of the gain corresponds to a better effect of split. XGBoost finds the optimal feature to split based on the exact greedy algorithm on account of increasing efficiency [36]. The split stops when the gain is smaller than zero, and the structure of CART is then determined. The next CART is constructed by the same steps and regards the residual of the previous tree as its training target. All of the CARTs can then constitute the XGBoost model and give the final predictive result. A detail description of XGBoost algorithm was introduced in Appendix A.

#### 2.4.2. *K*-fold Cross Validation (CV)

The *k*-fold CV is commonly adopted to improve the ability of model generalization in the field of machine learning [37]. In a *k*-fold CV, the entire dataset is randomly divided into *k* disjoint subsets of almost the same size, i.e., *k* folds. Then use one fold as the testing dataset and the remaining *k* − 1 folds as the training dataset until all the folds have been used as the testing dataset sequentially. In the present study, 5-fold CV was employed.

#### 2.4.3. Dealing with Class-Imbalance Data in Short-Term Prediction

In order to evaluate the predictive ability of hypnopompic HRV to short-term CVD outcomes, only 70 participants were included as the short-term CVD group because they had an occurrence of a cardiovascular event within two years after baseline. However, there were 998 participants in the non-CVD group, leading to an extremely imbalanced data for the prediction model. The performance of the classifiers could be, thus, influenced seriously because learning algorithms tend to predict all the minority class as the majority class to acquire higher accuracy [38]. In this study, we under-sampled the non-CVD group to match the short-term CVD group with equal sample size. Test of distribution similarity [39,40] between the original and under-sampled dataset for the non-CVD group was conducted to ensure the selected subset can generally represent for the whole non-CVD group.

#### 2.4.4. Performance Evaluation of Predictive Models

The analysis of the confusion matrix is the most essential and intuitive approach to estimate the performance of a classifier. For a multi-classification problem with m classes, the confusion matrix is defined as a m × m matrix, with its element $m_{ij}$ represented for the number of samples which belong to the *i*-th class, but are predicted as the *j*-th class. In a typical binary classifier, there are two classes, called “Positive” and “Negative”, respectively. The resulting four elements of its confusion matrix are, thus, corresponded to “True Positive” (TP), “False Negative” (FN), “False Positive” (FP) and “True Negative” (TN) [41,42]. In this study, the CVD group was considered as the Positive class and the non-CVD group as the Negative class, respectively. Accordingly, six measures for model performance, i.e., Accuracy (ACC), Sensitivity or Recall (TPR), Specificity (TNR), Precision (PPV), F1-Score (F1) and Matthew correlation coefficient (MCC), were calculated in the present study followed by Formula (1–6) [42].
(1)$$\mathrm{ACC}=\frac{\mathrm{TP}+\mathrm{TN}}{\mathrm{TP}+\mathrm{TN}+\mathrm{FP}+\mathrm{FN}}$$
(2)$$\mathrm{TPR}=\frac{\mathrm{TP}}{\mathrm{TP}+\mathrm{FN}}$$
(3)$$\mathrm{TNR}=\frac{\mathrm{TN}}{\mathrm{TN}+\mathrm{FP}}$$
(4)$$\mathrm{PPV}=\frac{\mathrm{TP}}{\mathrm{TP}+\mathrm{FP}}$$
(5)$$F1=\frac{2\times \mathrm{PPV}\times \mathrm{TPR}}{\mathrm{PPV}+\mathrm{TPR}}$$
(6)$$\mathrm{MCC}=\frac{\mathrm{TP}\times \mathrm{TN}-\mathrm{FP}\times \mathrm{FN}}{\sqrt{\left(\mathrm{TP}+\mathrm{FP}\right)\left(\mathrm{TP}+\mathrm{FN}\right)\left(\mathrm{TN}+\mathrm{FP}\right)\left(\mathrm{TN}+\mathrm{FN}\right)}}$$

### 2.5. Statistical Analyses

Statistical analyses were performed using MATLAB (Mathworks Inc., Natick, MA) and SPSS version 22 (IBM SPSS Statistics, NY, United States). Firstly, the difference in clinical characteristics between the CVD and non-CVD groups was assessed. Chi-square test was applied to categorical variables, such as gender, smoking status, diabetes and hypertension. Meanwhile, for the continuous variables, such as age, waist/hip ratio, BMI, height, lifetime cigarette smoking, AHI and RDI, the Lilliefors test was used to check the normality of distribution [43] at first. Then non-parametric Whitney test was used for the between-group difference if they violated the normality, otherwise t-test was adopted. The between-group difference in HRV metrics was evaluated in a similar way. Secondly, in order to test whether the under-sampled subset of the non-CVD group was representative, on the one hand, the Kolmogorov-Smirnov (K-S) test [39] was employed between the subset and the whole non-CVD group. On the other hand, Jensen-Shannon divergence (JSD) was calculated for the similarity of data distribution [40]. A value of JSD approaching to zero represents for substantial similarity between two distributions.

## 3. Results

### 3.1. Clinical Characteristics

The baseline clinical characteristics of the participants were reported in different forms in Table 2. Categorical variables, such as gender, diabetes and hypertension, were represented by the percentage of participants of each category. Others were expressed as mean ± standard deviation (SD) if they normally distributed otherwise as median [lower quartile, upper quartile]. 

As shown in Table 2, the CVD group had significantly greater age, percentage of males, BMI, height, waist/hip ratio, the morbidity of diabetes or hypertension, and sleep-related breathing disorders represented by AHI and RDI than the non-CVD group.

### 3.2. HRV Metrics

In Table 3, we illustrated the baseline HRV metrics of all the participants. Compared to the non-CVD group, CVD group was associated with a decrease in HRV metrics of frequency and time domain, especially the values of LF (p=0.004) and HF (p=0.004). As for the nonlinear HRV metrics, MSE10 (value of MSE under scale 10, p=0.018) and MPE (p=0.004) of CVD group were significantly higher than that of non-CVD group, while MSE1 (value of MSE under scale 1, p=0.005) and MSE2 (value of MSE under scale 2, p=0.037) were significantly lower in CVD group. There is no significant difference between the two groups of MSE under other scales.

### 3.3. Prediction of CVD Outcomes Based on XGBoost

We investigated the long-term and short-term predictive ability of XGBoost models, respectively. In both models, the feature vector for each participant consisted of 21 metrics, including 11 clinical characteristics (age, gender, BMI, height, waist/hip ratio, smoking status, lifetime cigarette smoke, diabetes, hypertension, AHI and RDI) and 10 hypnopompic HRV metrics (TP, LF, HF, Hfnorm, SDNN, RMSSD, MSE1, MSE2, MSE10 and MPE). All the participants, i.e., 1219 participants in the CVD group and 998 participants in non-CVD group were involved in the long-term prediction. As far as the short-term prediction is concerned, only 70 participants in the short-term CVD group and 70 participant under-sampled from the non-CVD group were included.

#### 3.3.1. Results of Distribution Similarity Tests for Under-Sampling

For each metric included in the feature vector of the short-term prediction model, the probability distribution of the under-sampled subset and the whole non-CVD group were illustrated in Figure 1. The results shown in Figure 1 demonstrated that all the used metrics had similar distributions in the under-sampled and the original non-CVD group, indicating the under-sampled non-CVD group was a representative subset of the original dataset.

#### 3.3.2. Performance of CVD Outcomes Prediction

The results of 5-fold CV of the two prediction models were illustrated in Table 4. There was a promotion of performance in the short-term prediction model compared with the long-term one. Compared with long-term prediction, ACC was improved relatively (calculated by growth value from long-term to short-term, then divided by the long-term value) by 10.7%, TNR by 34.8%, PPV by 13.7%, F1-scores by 4.2% and MCC by 39.1% in short-term prediction on average.

#### 3.3.3. Independent Predictive Ability of Hypnopompic HRV

For the purpose of exploring the independent predictive ability of hypnopompic HRV, we compared the performance of prediction models utilizing features of clinical characteristics with and without HRV metrics. As shown in Table 5, for the long-term prediction, an ACC of 73.5% and F1-score of 77.8% was achieved by the model fed with both clinical characteristics and HRV metrics. There was a slight decline of ACC, F1-score and MCC when without using HRV metrics. However, the performance of the short-term prediction without HRV metrics became much worse with a 6.1% relative decrease in ACC, a 3.8% relative dropdown in F1-score and a 14.1% relative decrease in MCC, compared to the one with HRV metrics, indicating a promising application of hypnopompic HRV in short-term prediction of CVD outcomes.

#### 3.3.4. The Importance of Features in Prediction Models

We further investigated the importance of the used features, i.e., the clinical characteristics and the HRV metrics, in the prediction models. For a XGBoost algorithm-based classifier, the importance of a feature is often assessed by the total times when it is employed to split a node of CARTs [44]. In this way, the importance of each feature in both long-term and short-term prediction models was illustrated in Figure 2. Although in both models, age and waist/hip ratio were two most important features, HRV metrics, especially the complexity measures, i.e., MSE2, MSE1 and MPE, also contributed substantially to the model performance. As shown in Figure 2, the HRV metrics ranked ahead in the short-term model compared to the long-term model. Moreover, HRV measures occupied 69.1% times in the splitting of CARTS among all the features except for the two most important features, i.e., age and waist/hip ratio, in the short-term model. However, this value decreases to 62.4% in the long-term model, suggesting a more important role of HRV metrics in the short-term CVD prediction. 

## 4. Discussion

In this study, conventional time domain and frequency domain analysis, as well as entropy-based nonlinear analysis of hypnopompic HRV were used to characterize the heart rate dynamics in baseline CVD-free participants. Machine learning models, which was fed with HRV metrics and some typical risk factors and further monitored by novel XGBoost algorithm, were constructed and evaluated for both long-term and short-term prediction of CVD outcomes. By comparing the performances of the model with and without HRV metrics, we verified the real contribution of HRV metrics in the prediction. Hypnopompic HRV was found to be assistant of the typical CVD risk factors, such as age and waist/hip ratio, to predict CVD outcomes, especially to predict the occurrence of cardiovascular events within two years. Moreover, entropy-based metrics of heart rate dynamics were demonstrated to have better predictive power compared with traditional measures of HRV according to their greater feature importance in the XGBoost models.

The current study is in consistence with our previous work [26], where age and waist/hip ratio were found of vital importance in the prediction of CVD outcomes. In the existing literature, aging was associated with cardiac dysfunction [45,46], and waist/hip ratio, which was generally regarded as a measure of obesity, was found to have a strong correlation with CVD [47]. Furthermore, the findings here demonstrate that complexity-based metrics of heart rate dynamics have better predictive power compared with traditional measures of HRV, supporting the view that entropy measures should be more sensitive as a reliable approach for ECG-based studies [48].

Compared with conventional linear HRV measures, entropy-based measures performed better in predicting CVD in this study, which might attribute to the intrinsic nonlinearity of heart rate dynamics. Such a nonlinearity would not only exert limitations on the use of traditional linear analyses [49,50], but also demand methods which can evaluate the underlying physiological process more suitably. MSE is widely used in coping with HRV signals as physiological controls exist from subcellular to systemic levels in the heart rate control system [30], while the MPE algorithm takes the advantages of symbolic dynamics analysis to be a robust approach for the evaluation of HRV [51,52].

It has been well-established that cardiovascular diseases are accompanied by an ANS imbalance with increased sympathetic or reduced vagal activity. HRV indices are commonly used markers for ANS function. Altered HRV was found in patients with myocardial infarction [53], diabetic [54] or CHF [55] when compared with healthy individuals. In the context of CHF, decreased HRV was associated with disease severity [56,57,58,59,60]. Moreover, by using nonlinear techniques to automatically classify the HRV signals of healthy people and those at risk of sudden cardiac death (SCD), Fujita [61] demonstrated that HRV was able to identify the person at four minutes prior to SCD. Carney [62] recently showed that high heart rate and low HRV during nighttime could be used as a predictor of the poor response to the treatment of major depression and of the higher risk for mortality in patients with stable coronary heart disease. This study further clarifies the association between sleep HRV and CVD, and establishes the role of hypnopompic HRV in the automatic prediction of short-term CVD events. The alteration of hypnopompic HRV is, thus, considered as a potential predictor for CVD events.

In this study, the application of machine learning in CVD outcomes is showing its promising in the early prediction of CVD outcomes. With the development of data-capture technologies, machine learning satisfies the demands of coping with sophisticated biomedical data. It can also integrate the information generated by plenty and heterogeneous existing modalities to describe physiological and neurobiological phenomena [63]. Numerous novel algorithms of machine learning have been proposed in recent years. Although the algorithm of machine learning is not the focus of the present work, the adopted XGBoost method is still a state-of-art approach recently [64]. Future work is encouraged to the application of more machine learning algorithms, such as deep learning in this field.

## 5. Conclusions

Measures of hypnopompic HRV can be a potential predictor for CVD events. Compared to conventional HRV time and frequency domain measures, entropy-based metrics from nonlinear complexity analysis can be more sensitive to predict the occurrence of CVD, thus, should be considered for CVD-free people in their long-term management and evaluations. Further prospective studies are encouraged to evaluate the proposed model for short-term prediction of CVD in a broader range of population and to apply entropy-based measures in the early detection of CVD risks.

## Figures and Tables

**Figure 1 entropy-22-00241-f001:**
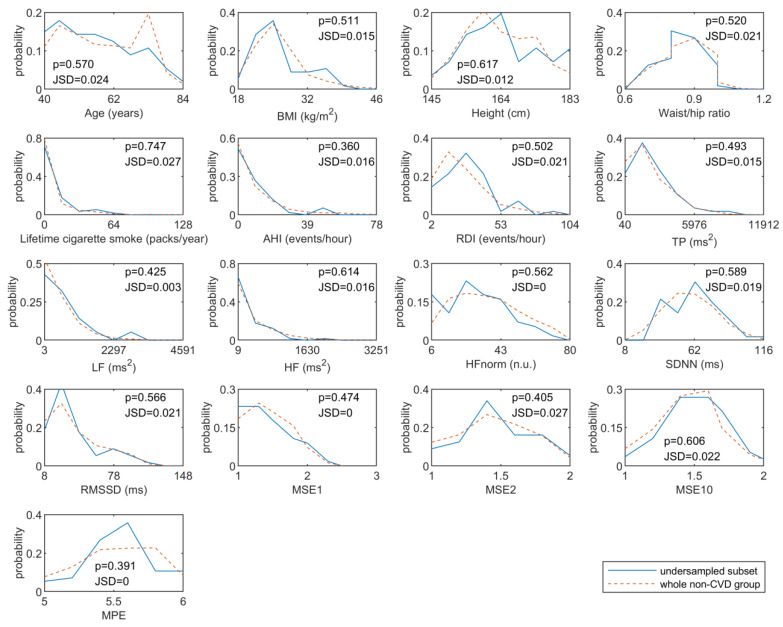
The results of distribution similarity tests between the under-sampled and the original non-CVD group. Logistic or classified features, including gender, smoking status, hypertension and diabetes, were excluded. In K-S test, a
p value less than 0.05 represents for a significant difference of distribution, while a value of JSD closing to one corresponds to a significant difference of distribution. K-S test = Kolmogorov-Smirnov test; JSD = Jensen-Shannon divergence.

**Figure 2 entropy-22-00241-f002:**
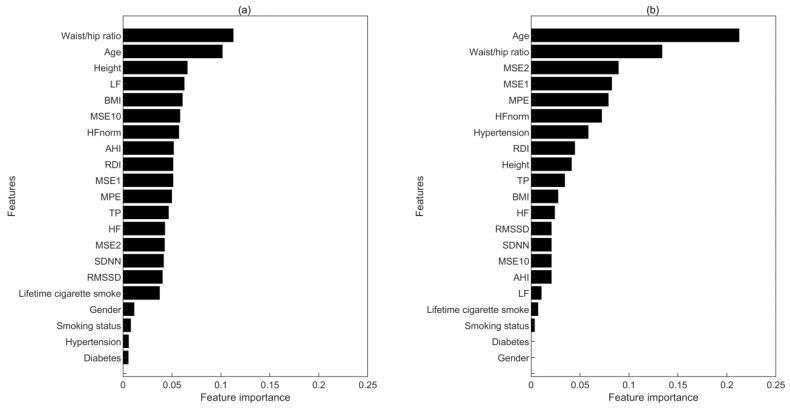
Feature importance in (**a**) long-term model and (**b**) short-term model. The horizontal axis shows the relative feature importance (i.e., the ratio of used times of each feature to the total used times of all features).

**Table 1 entropy-22-00241-t001:** Heart rate variability (HRV) metrics used in this study.

Usage	Metric	Units	Description
on each 5-min HRV segment	TP	ms^2^	Total power in frequency range (0.003–0.4 Hz) [12].
	LF	ms^2^	Power in low frequency range (0.04–0.15 Hz) [12]
	HF	ms^2^	Power in high frequency range (0.15–0.4 Hz) [12].
	HFnorm	n.u.	HF power in normalized units (HF/(LF + HF) × 100) [12].
on entire 1-h HRV data	SDNN	ms	Standard deviation of all RR intervals [12].
	RMSSD	ms	The square root of the mean of the sum of squares of differences between adjacent RR intervals [12].
	MSE		Multiscale sample entropy [30] of the RR intervals on 10 time scales. To calculate the sample entropy on each scale (denoted as MSE1, MSE2, …, MSE10 sequentially), the embedding dimension was set as 2 and the tolerance as 0.15 × SD, where SD is the standard deviation of the original time series.
	MPE		Modified permutation entropy of RR intervals [31,32], with an embedding dimension value of 4.

**Table 2 entropy-22-00241-t002:** Baseline clinical characteristics of participants involved in this study.

	CVD	non-CVD	*p*
Number of participants	1219	998	
Age (years)	63[58,69]	60[50,73]	<0.001 *
Gender (Male/Female,%)	47.3/52.7	39.2/60.8	<0.001 *
BMI (kg/m2)	28.2[25.4,31.3]	27.1[24.4,30.4]	<0.001 *
Height (cm)	167[160,175]	165[158.8,174]	<0.001 *
Waist/hip ratio	95.1[90.1,99.2]	89.9[81.5,96.2]	<0.001 *
Smoking status (Never/Current/Former,%)	49.9/7.3/42.8	54.6/7.1/38.3	0.023 *
Lifetime cigarette smoke (packs/year)	0[0,19]	0[0,12]	0.014 *
Diabetes (Yes/No,%)	7.2/92.8	3.3/96.7	<0.001 *
Hypertension (Yes/No,%)	41.5/58.5	33.6/66.4	<0.001 *
AHI (events/hour)	9.9[4.2,19.1]	8.3[3.3,16.9]	0.025 *
RDI (events/hour)	30.3[19.2,45]	26.9[17.1,40.1]	<0.001 *

Note: Values are reported as number and percent, or as median [lower quartile, upper quartile]. BMI = body mass index; AHI = apnea hypopnea index; RDI = respiratory disturbance index. * represents a significant difference, *p* < 0.05 ($\chi^{2}$ test, t-test or non-parametric test).

**Table 3 entropy-22-00241-t003:** HRV metrics of participants involved in this study.

	CVD	non-CVD	*p*
TP(ms^2^)	2299.4[1458.6,3410.9]	2324.5[1412.7,3802.3]	0.186
LF(ms^2^)	496.4[296,807.1]	528.1282.7,929.9]	0.004 *
HF(ms^2^)	251.3[120.2,596.9]	308.3[132,707.8]	0.004 *
HFnorm(n.u.)	35.9[24.2,50.8]	38.2[25.4,52.1]	0.188
SDNN(ms)	63.5[52.4,76.5]	64.8[51.7,79.1]	0.18
RMSSD(ms)	36.3[25.1,60.3]	39.4[26.3,62.1]	0.137
MSE1	1.41[1.14,1.71]	1.49[1.21,1.78]	0.005 *
MSE2	1.47[1.25,1.69]	1.51[1.3,1.73]	0.037 *
MSE3	1.46[1.28,1.64]	1.49[1.28,1.66]	0.062
MSE4	1.49[1.32,1.64]	1.48[1.3,1.66]	0.941
MSE5	1.53[1.38,1.68]	1.54[1.36,1.69]	0.979
MSE6	1.57[1.4,1.73]	1.57[1.39,1.7]	0.465
MSE7	1.57[1.41,1.73]	1.58[1.4,1.72]	0.543
MSE8	1.58[1.42,1.74]	1.57[1.4,1.71]	0.068
MSE9	1.58[1.4,1.74]	1.56[1.4,1.71]	0.06
MSE10	1.57[1.41,1.73]	1.55[1.39,1.7]	0.018 *
MPE	5.69[5.49,5.84]	5.61[5.41,5.82]	<0.001 *

Note: Values are reported as median [lower quartile, upper quartile]. * represents a significant difference, *p* < 0.05 ($\chi^{2}$ test, t-test or non-parametric test).

**Table 4 entropy-22-00241-t004:** The performance of the long-term prediction model and short-term prediction model.

		ACC (%)	TPR (%)	TNR (%)	PPV (%)	F1 (%)	MCC
long-term	1-fold	69.7	80.2	56.8	69.4	74.4	0.38
	2-fold	74.4	84.8	61.8	73.0	78.5	0.48
	3-fold	72.6	87.7	54.3	70.1	77.9	0.45
	4-fold	73.8	80.7	65.3	74.0	77.2	0.47
	5-fold	77.1	87.9	63.9	74.8	80.8	0.54
	average	73.5	84.2	60.4	72.3	77.8	0.46
short-term	1-fold	82.1	78.6	85.7	84.6	81.5	0.64
	2-fold	75.0	64.3	85.7	81.8	72.0	0.51
	3-fold	85.7	100.0	71.4	71.4	87.5	0.75
	4-fold	85.7	78.6	92.9	91.7	84.6	0.72
	5-fold	78.6	85.7	71.4	75.0	80.0	0.58
	average	81.4	81.4	81.4	82.2	81.1	0.64

Note: ACC = Accuracy; TPR = Sensitivity or Recall; TNR = Specificity; PPV = Precision; F1 = F1-Score; MCC = Matthew correlation coefficient.

**Table 5 entropy-22-00241-t005:** The performance of two prediction models consisting of different feature vectors.

Prediction Model	Components of Feature Vector	ACC (%)	TPR (%)	TNR (%)	PPV (%)	F1 (%)	MCC
long-term	clinical characteristics and HRV metrics	73.5	84.2	60.4	72.3	77.8	0.46
	only clinical characteristics	72.9	82.4	61.3	72.3	77.0	0.45
short-term	clinical characteristics and HRV metrics	81.4	81.4	81.4	82.2	81.1	0.64
	only clinical characteristics	76.4	85.7	67.1	72.8	78.4	0.55

Note: All models were based on 5-fold cross validation, but only reported the average value of each metrics in this table. ACC = Accuracy; TPR = Sensitivity or Recall; TNR = Specificity; PPV = Precision; F1 = F1-Score; MCC = Matthew correlation coefficient.

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
