# Peer review of "Entropy-Based Measures of Hypnopompic Heart Rate Variability Contribute to the Automatic Prediction of Cardiovascular Events"

_entropy, 2020, doi:10.3390/e22020241_

Round 1

Reviewer 1 Report

The authors used HRV metrics and typical CVD risk factors as input features of machine learning to improve the prediction of CVD. The results are positive.

I have several minor questions:

In lines 22-23, the authors stated that “Compared to conventional HRV measures, entropy-based metrics is more sensitive in screening CVD.” However, Table 5 lacked the comparison between “only clinical characteristics” and “only HRV metrics”. Thus, I suggest to change that statement, or to compare the two cases with each other.

In line 68, can the authors briefly explain what CVD events are included and according to what standard? It would be helpful for readers who are not familiar with medicine.

In line 110, the authors stated that “conventional frequency-domain analysis was employed on each 5-minute HRV segment”. Does that mean there were 12 segments for each subjects? Or, only the first 5-minute HRV segment was extracted?

Were the numbers correct in lines 238-239 and lines 248-249?

In Figure 2, was the fifth parameter PE or MPE?

Reviewer 2 Report

ENTROPY-BASED MEASURES OF HYPNOPOMPIC HEART RATE VARIABILITY CONTRIBUTE TO THE AUTOMATIC PREDICTION OF CARDIOVASCULAR EVENTS

The hypothesis of this study is that entropy-based measures of hypnopompic heart rate variability (HRV) could be important to predict cardiovascular disease (CVD).
The Authors used an open access dataset of polysomnography signals. They included 2442 participants who where CVD-free at the beginning of the study and divided them in 2 groups:

  1. 1)  non-CVD group (they hadn’t CVD incidents during the follow up)

  2. 2)  CVD group (they had CVD incidents during the follow up)

Authors used machine learning (XGBoost) to predict the probability of CVD incidents in short and long term, using HRV and other clinical characteristics (age, gender, height, waist/hip ratio, ecc). In particular they used complex HRV indicators based on entropy measures.

The achieved predictive performance was good. In particular: the model performs better in short-term prediction and the most important features are: age, waist/hip ratio, MSE2 and MSE1. They demonstrated that HRV entropy-based measures are important to predict CVD incidents both in short and long term.

Comment

There is a clear background and the method is well defined. Moreover, the result is clearly described and the model prediction is reliable.
The abstract is complete and clear, it shows all the main points of the article.
The main question is relevant and interesting because CVD is one of the major causes of mortality and the possibility to predict the incidence of these diseases is worth investigation. The machine learning method is accurate.

Confound variables were well controlled by the division of participants in two groups (non-CVD group, the control group and CVD-group, the experimental group) and the exclusion of participants whose RR intervals were too disturbed.​ The HVR analysis, the analysis of confusion matrix, the statistical analysis and the machine learning model are well defined.

However, the methodology and research purpose is not novel. The study does not add a strong scientific/methodological contribution, as applications of machine learning on HRV data to predict clinical conditions are a well covered topic in past literature.

The relevant contributions of this study are:

  • -  Authors tried the model with and without HRV metrics in order to verify their real

    contribution in the prediction;

  • -  They aimed at demonstrating that entropy-based metrics of heart rate dynamics have

    better predictive power compared with traditional measures of HRV;

  • -  The use of the (quite) novel XDGBoost predictive models.

    I think that the relevance of these contributions should be highlighted more.
    The study might be of interest for publication in this journal given the use entropy indicators and their importance in the prediction. But it would then seem critical to comment more on

the physiological relevance of entropy indicators. E.g.: Why are they important? What are the physiological characteristics that are “encoded” into entropy indicators? Why are entropy-based indicators better than others?

In addition there are some minor issues that should be addressed:

  1. The Authors use the term “obvious*” in four sentences (lines 19, 205, 237, 263), but

    the reader might find the sentences not obvious at all.. I would suggest to remove the term and, instead, add a sentence to explain why the discussed aspects were expected.

  2. A “one fit for all” classification metrics could be used: the Matthew correlation coefficient, which is robust to class imbalance and better quantifies the confusion matrix patterns.

  3. It is not clear how the Authors managed the frequency domain indexes on the multiple 5-mins portions: was the average of all the portions used for classification?

I suggest that the paper undergoes a minor revision before it is ready for publication.
